# Global Prevalence and Hemagglutinin Evolution of H7N9 Avian Influenza Viruses from 2013 to 2022

**DOI:** 10.3390/v15112214

**Published:** 2023-11-04

**Authors:** Qianshuo Liu, Haowen Zeng, Xinghui Wu, Xuelian Yang, Guiqin Wang

**Affiliations:** 1College of Life Sciences, Nanjing Normal University, Nanjing 210023, China; 211202116@njnu.edu.cn (Q.L.); zenghw8311@163.com (H.Z.); w63372011@163.com (X.W.); 2Nanjing Advanced Academy of Life and Health, Nanjing 211135, China; xlyang@ipsnanjing.cn

**Keywords:** H7N9 avian influenza viruses, epidemiology, genetic evolution, selective pressure, glycosylation site

## Abstract

H7N9 avian influenza viruses have caused severe harm to the global aquaculture industry and human health. For further understanding of the characteristics of prevalence and hemagglutinin evolution of H7N9 avian influenza viruses, we generated the global epidemic map of H7N9 viruses from 2013 to 2022, constructed a phylogenetic tree, predicted the glycosylation sites and compared the selection pressure of the hemagglutinin. The results showed that although H7N9 avian influenza appeared sporadically in other regions worldwide, China had concentrated outbreaks from 2013 to 2017. The hemagglutinin genes were classified into six distinct lineages: A, B, C, D, E and F. After 2019, H7N9 viruses from the lineages B, E and F persisted, with the lineage B being the dominant. The hemagglutinin of highly pathogenic viruses in the B lineage has an additional predicted glycosylation site, which may account for their persistent pandemic, and is under more positive selection pressure. The most recent ancestor of the H7N9 avian influenza viruses originated in September 1991. The continuous evolution of hemagglutinin has led to an increase in virus pathogenicity in both poultry and humans, and sustained human-to-human transmission. This study provides a theoretical basis for better prediction and control of H7N9 avian influenza.

## 1. Introduction

The novel H7N9 avian influenza viruses [1], which commenced infecting humans in early 2013 [2], were a result of the recombination of the hemagglutinin (HA) of the H7N7 avian influenza viruses, the neuraminidase (NA) of the H7N9 viruses, and the internal gene segments of H9N2 viruses. In 2016, following the introduction of novel H7N9 viruses from the Yangtze River Delta to the Pearl River Delta region, an insertion mutation of four amino acids, RKRT, occurred at the cleavage site of the HA protein. This mutation led to an elevation in the virulence of the viruses in avian species, subsequently resulting in outbreaks across eight provinces in China [3]. The highly pathogenic (HP) H7N9 viruses caused almost complete mortality in infected chickens, leading to significant poultry losses and culling. Moreover, these viruses also inflicted a mortality rate of approximately 50% among human cases [4]. From its initial outbreak until October 2017, the novel H7N9 avian influenza viruses had caused five major epidemics, with the World Health Organization (WHO) reporting a total of 1564 laboratory-confirmed cases of human infection [5]. Thereafter, the Chinese government initiated large-scale mandatory immunization programs for poultry, which proved to be effective in reducing the incidence of avian infection and reducing human infections. During the 2018 flu season, only three cases of human infection were reported, and as of May of that year, there were only five reported incidents of avian outbreaks [6]. However, despite the implementation of mandatory immunization, the novel H7N9 viruses were not eradicated. In 2018 and 2019, these viruses infected poultry again in chicken farms and zoos in northern China [7]. In late March 2019, after over a year of no reported cases, human infections with the HPAI H7N9 viruses resurfaced [8].

In addition to the new H7N9 virus that emerged in 2013, the earlier H7N9 avian influenza viruses have been reported in wild birds in the USA, Canada, Guatemala, Spain, Sweden, Egypt, Mongolia, and Taiwan [9]. In recent years, the earlier H7N9 influenza viruses have been detected multiple times in poultry and wild birds in various states in the Midwest of the United States [10], including both low-pathogenicity strains and highly pathogenic ones [11,12]. The origins of the earlier H7N9 viruses in Asia can be traced back to the year of 2000 [13]. In 2021, a low-pathogenicity strain of the earlier H7N9 avian influenza viruses was isolated in South Korea [14]. H7N9 avian influenza viruses pose a threat to global poultry farming and human health [15].

HA serves as one of the key membrane proteins of avian influenza A viruses [16]. The HA protein is responsible for binding the viruses to sialic acid receptors on the surface of host cells, and is one of the factors that determines the host species [17,18]. The HA protein is also an important target protein for humoral immunity of influenza virus. The HA plays a critical role in the transmission, evolution, immune response and lifecycle of avian influenza viruses [19,20]. Understanding the genetic evolution characteristics of HA is crucial in preventing and controlling avian influenza pandemics.

To gain insights into the global prevalence and HA evolution characteristics of H7N9 viruses, this study conducted a comprehensive analysis of the global circulation of H7N9 viruses from January 2013 to December 2022. This study involved constructing phylogenetic trees and lineage assignment of the HA protein of these viruses. Our study predicted the N-glycan positions on the HA protein for the virus strains that have continued to circulate since 2019. Furthermore, this study analyzed and compared the selective pressures acting on the HA protein.

## 2. Materials and Methods

### 2.1. Data Collection and Compilation

For this study, a total of 2597 H7N9 avian influenza virus strains were collected from the Global Initiative on Sharing All Influenza Data (GISAID) database. The dataset included information such as the date and location of sample collection, as well as the genetic and protein sequences of 2549 virus strains [21]. The study period spanned from 1 January 2013, to 31 December 2022, excluding any virus strains labeled as “reassortant” in their names. Based on the temporal and spatial data, a bar chart was created using the ggplot2 package in the R programming language [22]. Additionally, a global map depicting the prevalence of H7N9 avian influenza was generated using the Datawrapper website (https://www.datawrapper.de/ accessed on 15 March 2023). For detailed information on virus sample collection dates, locations, and other relevant details, please refer to Appendix A.

### 2.2. Construction of the Phylogenetic Tree

In this study, the BioAider V1.527 software [23] was utilized for the alignment of gene sequences using the MAFFT algorithm [24]. The aligned gene sequences were subsequently trimmed in the MEGA 11 software, retaining only the coding region between the start and stop codons. For reducing computational load, the trimmed gene sequences were subjected to 100% deduplication using the BioAider V1.527 software [23]. The neighbor-joining (NJ) method in MEGA 11 software was employed to construct the phylogenetic tree [25,26]. The resulting tree was further refined and annotated using the FigTree 1.4.4 software, which facilitated the labeling of virus lineages, sample collection locations, and dates (http://tree.bio.ed.ac.uk/software/Figtree/, accessed on 20 January 2023). The non-redundant HA gene sequences can be found in Appendix A.

We also downloaded all the HA sequences of H7 subtype influenza viruses collected from 2013 to 2022 from the GISAID database. A phylogenetic tree was constructed with the same method. We used MEGA software to calculate the mean genetic distance of H7, H7N9, and the novel H7N9 avian influenza viruses [25].

### 2.3. The Estimation of the Time to the Most Recent Common Ancestor (tMRCA)

The HA gene sequences of the viruses, the phylogenetic tree, and the sample collection dates of the viruses were imported into the Treetime V 0.11.1 program. Utilizing the linear relationship between the root-to-tip genetic distance and the sample collection dates of virus strains, we estimated, respectively, the time to the most recent common ancestor (tMRCA) for the HA genes from the H7N9 influenza viruses and the H7N9 viruses in the evolutionary tree [27].

### 2.4. Prediction of N-Glycan Positions on the HA Protein of H7N9 Avian Influenza Viruses since 2019

The amino acid sequences of the HA protein of the H7N9 viruses after 2019 were input into the online server NetNGlyc 1.0 to predict potential N-glycan positions [28]. The server predicted potential sites based on the presence of the N-X-S/T motif on HA proteins, where X represents any amino acid except proline. A threshold value of >0.5 suggests glycosylation [29]. The amino acid lollipop plot, which described the N-glycosylation sites, was generated using the TrackViewer V 1.36.2 package in R [30].

### 2.5. Analysis of the Selection Pressure of H7N9 Viruses HA since 2019

The selective pressure analysis and identification of positively selected sites for different lineages of H7N9 viruses were conducted, using the online server Datamonkey [31], after removing stop codons and signal peptides from the aligned HA gene sequences. The Omega value (dn/ds), which represents the ratio of the non-synonymous substitution rate (dn) to the synonymous substitution rate (ds), is used to measure the selection pressure. A dn/ds > 1 indicates positive selection, <1 indicates negative selection, and =1 indicates neutral selection [32].

In order to identify positively selected sites, four different algorithms were utilized: Fixed Likelihood Ancestor Counting (FLAC) [33], Mixed-Effects Model of Evolution (MEME) [34], Fixed Effects Likelihood (FEL) [33], and Fast Unconstrained Bayesian Approximation (FUBAR) [35]. The thresholds for the calculation results of MEME, FLAC, and FEL were set at *p* < 0.05, while the threshold for FUBAR was set at *p* > 0.9. Only when at least two algorithms supported the positive selection at specific sites were those sites considered positively selected.

Furthermore, the dn/ds ratios, reflecting the selection pressure, was calculated separately for the HA genes of different lineages of H7N9 avian influenza viruses, from the different virus hosts, using the MEME and SLAC algorithms. The average values were determined using GraphPad Prism software (version 9.1.0, GraphPad Software, San Diego, CA, USA), and a graph depicting the dn/ds ratios of different lineages was generated.

## 3. Results

### 3.1. Global Prevalence of H7N9 Avian Influenza Viruses

Though H7N9 avian influenza also appeared sporadically in other regions worldwide, China has witnessed periodic outbreaks from 2013 to 2017. Since 2017, its outbreaks have transitioned into a more sporadic pattern (Figure 1A), as reported in Asia (Chinese mainland, Taiwan, Hong Kong, South Korea, Japan, Malaysia, and Bangladesh), Europe (the Netherlands and the United Kingdom), North America (Canada and the United States), Africa (Egypt), and Oceania (Australia). Among the regions affected by the viruses, Asia has the highest number of cases, with a significant concentration in the eastern region, particularly Chinese mainland (Figure 1B).

By 2018, H7N9 avian influenza had experienced five periodic outbreaks as follows: the first wave from February 2013 to September 2013, the second from October 2013 to September 2014, the third from October 2014 to September 2015, the fourth from October 2015 to September 2016, and the fifth from October 2016 to September 2017. These five major outbreaks primarily occurred in Chinese mainland. Since the middle of 2017, the number of H7N9 virus cases have started to decrease, although sporadic cases still occurred. Since March 2021, H7N9 viruses have not been isolated (Figure 1A).

### 3.2. Phylogenetic Analysis of HA Genes of H7N9 Avian Influenza Viruses

The phylogenetic tree was constructed using 1540 non-redundant HA gene sequences of the H7N9 avian influenza viruses from 2013 to 2022. Based on the topology of the tree, the location and time of virus isolation, the common ancestor and previous studies, the HA genes were classified into six distinct lineages labeled A, B, C, D, E, and F (Figure 2A).

The viruses belonging to the lineages A, B, C, and D were classified as the novel H7N9 avian viruses and were more abundant, while those in lineages E and F, were less numerous (Figure 2A).

In terms of geographic distribution, the H7N9 avian influenza viruses in the different lineages exhibit differences. The viruses in the lineages A, C, and D are primarily prevalent in Asia. Within the lineage B, most of the viruses are isolated in Asia, except for one isolated in North America. Notably, all the high-pathogenicity viruses found in Chinese mainland belong to the lineage B. The viruses in the lineage F are exclusively distributed in North America, while the viruses in the lineage E viruses were be found in Oceania, Africa, Europe, and Asia.

The HA genes from the different lineages exhibit differences in their evolutionary characteristics. Compared with other HA genes, the HA phylogenetic tree branches of the high-pathogenicity viruses in the lineage B and the viruses in the lineage F demonstrates longer branch lengths. That indicates more genetic divergence and low homology of HA genes within these lineages (Figure 2A). The branches of the HA phylogenetic tree of the low-pathogenicity viruses in the lineages A, B, C, and D display a comb-like structure, characterized by short horizontal distances and relatively larger number of virus strains horizontally. This suggests that the high homology of HA genes within these lineages (Figure 2A).

The predominant lineages of the H7N9 avian influenza viruses vary across different years. In 2013, the H7N9 viruses were primarily from the lineages C, D, and E. During the period of 2014 to 2015, the low-pathogenicity viruses was from the lineage B. In 2016, the viruses from the lineages A and B became the main reservoirs, with minor presence in the lineages C and D. From 2017 to 2018, the viruses from the lineages A and B remained predominant, accompanied by a smaller presence in the lineage D. In 2018, all viruses were isolated from the lineages A and B, while after 2019, only the viruses from the B lineages remained. Notably, the viruses from the lineage E exhibited continuous distribution in the evolutionary tree from 2013 to 2021, except for 2017 and 2019. The viruses from the lineage F were observed between 2015 and 2017, and subsequently reemerged in 2020 (Figure 2A).

We constructed the HA phylogenetic tree of the H7 viruses from 2013 to 2022 (Figure 2B). Then, we compared the genetic diversity of the HA sequences from all the H7, H7N9 and novel H7N9 viruses. The genetic diversity was measured by the mean genetic distance. The two have a positive correlation. The mean genetic distance of the HA sequences of all the H7, H7N9 and novel H7N9 viruses from 2013 to 2022 is 0.07, 0.03 and 0.02, respectively. That indicates the degree of HA gene diversity from high to low is the H7, H7N9 and novel H7N9 viruses.

### 3.3. Time to the Most Recent Common Ancestor (TMRCA)

Using the Treetime V 0.11.1 program, the linear relationship between the root-to-tip genetic distance of HA genes of the H7N9 avian influenza viruses and the date of virus sample collection was calculated. The results revealed that HA genes from the different lineages could be traced back to a common ancestor in September 1991 (Figure 3A). The. tMRCA of the HA genes of the novel H7N9 influenza viruses, is January 2013(Figure 3B).

### 3.4. Characteristics of Evolution, Prevalence and N-Glycans of H7N9 Avian Influenza Viruses after 2019

Since 2019, a total of 78 H7N9 viruses have been distributed among the lineages B, E, and F (Figure 2). Among them, the lineage B viruses originated from China, including 71 strains of both high and low pathogenicity. The lineage E viruses were found in South Korea, comprising three strains. The lineage F viruses were identified in the United States, consisting of four strains (Figure 4A,B).

The potential N-glycan positions were predicted based on the amino acid sequences of the HA protein of H7N9 viruses after 2019. High-pathogenicity viruses from the lineage B possess four glycosylation sites: 12, 28, 123, and 231, while low-pathogenicity viruses from the lineage B lack the site of 123. Viruses from the lineages E and F have three potential glycosylation sites: 12, 28, and 231. These results indicate that the lineages B, E, and F share the 12, 28, and 231 sites, while the 123 site is unique to the high-pathogenicity viruses in the lineage B (Figure 4C and Table 1). These potential glycosylation sites are located in the HA1 region of the HA protein.

### 3.5. The Selection Pressure of H7N9 Viruses HA since 2019

Utilizing the Datamonkey online server, the selection pressure of the specific sites of the HA sequences of the H7N9 viruses after 2019 were analyzed. The analyses revealed positive selection signals at amino acid positions 47, 253, 546, and 567 in the HA protein in the lineage B viruses. These variations may play a role in viruses’ adaptation to the current environment or other factors. No positive selection sites were detected in the remaining lineages, indicating that the HA protein of the lineage B may exhibit a higher antigenic drift rate.

After 2019, the viruses in the lineages E and F exclusively infected avian, and the lineage B viruses were from avian, humans, mammals and the environment. The Omega values (dn/ds) of the HA genes from the different hosts in the lineages B, E and F were compared. The results showed that the Omega values of HA genes of the B lineage viruses from environmental, avian, and mammalian were 0.499, 0.401, and 0.316, respectively. The Omega values of HA genes in the E and F lineage viruses were 0.401 and 0.316, respectively (Table 2). All the Omega values were less than 1, indicating negative selection pressure on the HA genes. These findings imply that HA genes in the lineages B, E and F tend to retain conserved traits after undergoing limited adaptive evolution. The higher Omega value indicates that the HA genes in the lineage B may experience a relatively moderate negative selection pressure. The dn/ds ratio of the HA genes in the B lineage viruses from human is lower than that from avian, indicating that the negative selection pressure on the HA genes from human is greater than that from avian. The HA genes in the lineages E and F exist at higher synonymous mutation rates and lower non-synonymous mutation rates, indicating that they have stronger negative selection pressure (Figure 5).

## 4. Discussion

The threat posed by H7N9 avian influenza viruses is escalating. Typically, the viruses require prolonged evolution to cause major epidemics and cross-species transmission [36]. According to this study, the common ancestor of H7N9 avian influenza viruses emerged in September 1991 (Figure 3A). In 2013, the novel H7N9 avian influenza viruses first crossed the species barrier and infected humans. In 2016, an insertion mutation occurred at the HA protein cleavage site, increasing the pathogenicity of the viruses in both poultry and humans [3].

The H7N9 avian influenza viruses have not acquired the ability for sustained human-to-human transmission [37]. It has been confirmed that although viruses from the lineage B were introduced to Canada through human air travel from China, local transmission has not occurred [38]. In 2017, co-circulation of lineages from the Yangtze River Delta and the Pearl River Delta regions in China increased the risk of recombination and mutation among viruses with different pathogenicity [39].

In 2020, a novel low-pathogenic H7N9 virus originating from poultry was transmitted across species, infecting non-human mammalians, such as camels [40]. The novel H7N9 viruses, originating in China, have undergone significant changes, with the main circulating lineages shifting the lineages C and D to lineages A and B. Since 2019, the lineage B has been predominant, with high-pathogenicity viruses dominating in this lineage. The H7N9 viruses in the lineages E and F remain relatively low in number, and there have been no major shifts in the predominant lineages (Figure 2A). However, the H7N9 viruses in the lineages E and F have also been involved in cross-species transmission events between poultry and wild animals [12].

The H7N9 avian influenza viruses in the lineages A, B, C and D, despite their different evolutionary patterns, continue to evolve. It is crucial for governments worldwide to actively implement epidemic prevention and control measures to prepare for the potential occurrence of a large-scale H7N9 avian influenza pandemic.

The accumulation of N-glycosylation sites is a common immune evasion strategy used by various enveloped viruses [41]. By adding glycans in the antibody recognition region, viruses can escape host immunity through steric hindrance [42]. The globular head of HA contains the receptor binding region and major antigenic sites, serving as the primary binding sites for neutralizing antibodies [43]. Glycans have the ability to conceal antigenic epitopes and impede their binding with antibodies, thereby altering the immunological pressure on the population or enhancing viral fitness [29]. This phenomenon has been corroborated in studies on H1N1 and H3N2 avian influenza viruses [44,45]. Upon initial infection with avian influenza viruses or vaccination, the majority of antibody responses are directed toward the HA head domain, which serves as the target of vaccines [46]. Alterations in glycosylation sites can impact the efficacy of vaccines targeting HA. Among the predicted glycosylation sites in this study, positions 123 and 231 are located in the head region of the HA protein, while positions 12 and 28 are in the stalk region [47]. Compared to the HAs in other lineages, the HAs in the lineage B, particularly that of the high-pathogenicity strains, exhibits an additional glycosylation site at position 123, which is associated with a higher number of high pathogenic viruses within the lineage B. Therefore, this glycosylation site may be one of the contributing factors to the sustained circulation of the H7N9 viruses in the lineage B. In contrast to the head region, the stalk region of the HA protein has lower immunogenicity, and glycosylation at these sites primarily affects the structure and oligomerization of HA [48], potentially without a considerable impact on viruses’ circulation.

Since 2019, all the H7N9 HAs worldwide have been experiencing purifying selection, indicated by dn/ds < 1. This suggests that non-synonymous mutations may contribute to the increased diversity in the natural selection gene pool compared to synonymous mutations [49]. The higher dn/ds values observed in the lineage B indicate a stronger positive selection pressure, further supporting the notion that the increased glycosylation sites in the HA head region enhance the viruses’ ability to adapt to new hosts [50]. Amino acid positions 47, 253, 546, and 567 in the lineage B viruses have undergone adaptive changes and are likely involved in the functional evolution of H7N9 viruses.

The novel H7N9 influenza viruses caused five waves of human infections from 2013 to 2017, with a total of 1568 cases, of which 615 were fatal [4]. The novel H7N9 influenza viruses were low pathogenicity in birds. In early 2017, some low-pathogenicity H7N9 strains became highly pathogenic in chickens, causing severe influenza outbreaks in poultry in China [4]. H7N9 viruses posed the severe threat to human and animal health. To control the epidemic, the poultry were mandatorily immunized with H7N9 inactivated vaccine from September 2017 in China [15]. After birds were inoculated with the H7 vaccine, infection cases in humans and birds dramatically reduced. This suggests that the national poultry vaccination strategy has successfully controlled H7N9 infections in poultry and humans. Compared to the earlier novel H7N9 viruses, the HA gene of the H7N9 virus from the latest human case showed a relatively long genetic distance [4]. This indicates that the H7N9 viruses have evolved considerably even though vaccination dramatically limited the H7N9 prevalence. H7N9 viruses will not be eradicated. The viruses lurk in natural hosts such as waterfowls and continue to evolve, occasionally infecting other animals such as poultry. Once the virus has accumulated enough mutations and recombination, it could lead to the next pandemic. Based on World Health Organization (WHO) recommendations, a total of six virus strains were used for the development of influenza A(H7N9) vaccines from 2013 to 2023. The WHO may continue to recommend vaccine strains until all cases of human and poultry infection have disappeared.

This study analyzed the global epidemiology and the HA evolutionary characteristics of the H7N9 viruses from 2013 to 2022. The Gisaid data may have a lag, and some viruses may be collected in 2022 but uploaded in 2023. These viruses will not be included in our study. This study also predicted the glycosylation sites of the HA protein in the different lineages of H7N9 strains since 2019. By comparing the selection pressure on the HA protein of H7N9 viruses among the different lineages, this study provided insights into HA evolutionary dynamics. In summary, this study provides a theoretical basis for better prediction and control the spread of H7N9 avian influenza.

## 5. Conclusions

H7N9 avian influenza appeared sporadically in other regions worldwide, but China witnessed periodic concentrated outbreaks from 2013 to 2017. HA genes of the H7N9 avian influenza viruses originated from a common ancestor in 1991 and have undergone multiple evolutions and mutations. The year 2013 witnessed its first cross-species infection of humans, while 2016 witnessed an insertion of a mutation in the cleavage site of the HA protein, which increased its pathogenicity. There are differences in the HA evolutionary pattern of H7N9 avian influenza viruses of the different lineages, with a higher number of glycosylation sites in highly pathogenic viruses of the B lineage, which may account for their persistent pandemic. Globally, the HA proteins of H7N9 viruses have undergone a natural selection and adaptive changes, and in particular, HA genes of the lineage B have been subjected to a strong positive selection pressure, which further promotes the viruses’ ability to adapt to new hosts.

## Figures and Tables

**Figure 1 viruses-15-02214-f001:**
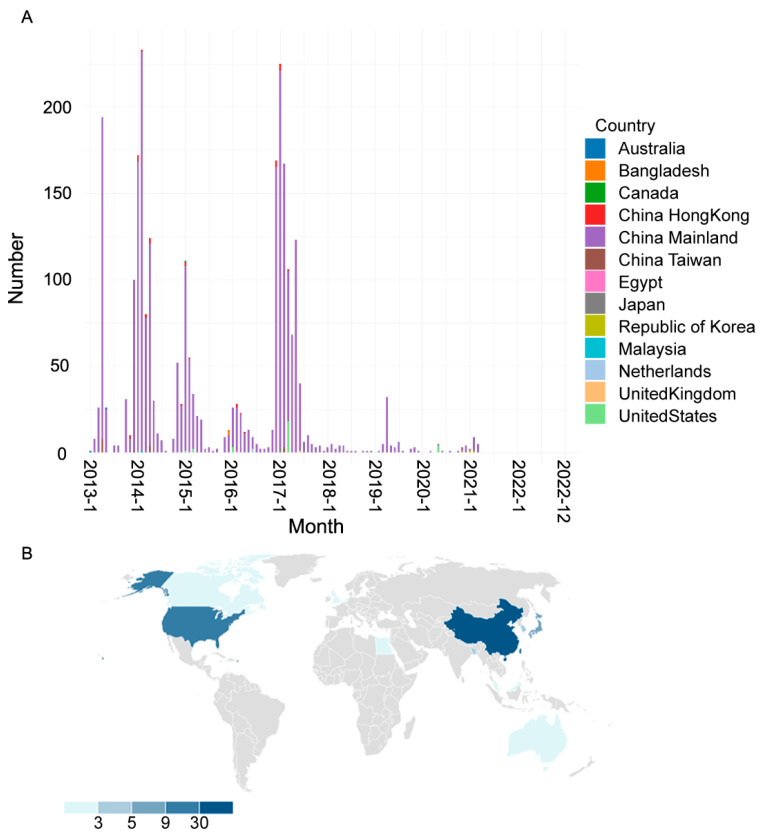
Global prevalence of H7N9 avian influenza from 2013 to 2022. (**A**) The distribution of H7N9 viruses in different countries and months. (**B**) The world map shows the total number of H7N9 virus strains isolated in different countries.

**Figure 2 viruses-15-02214-f002:**
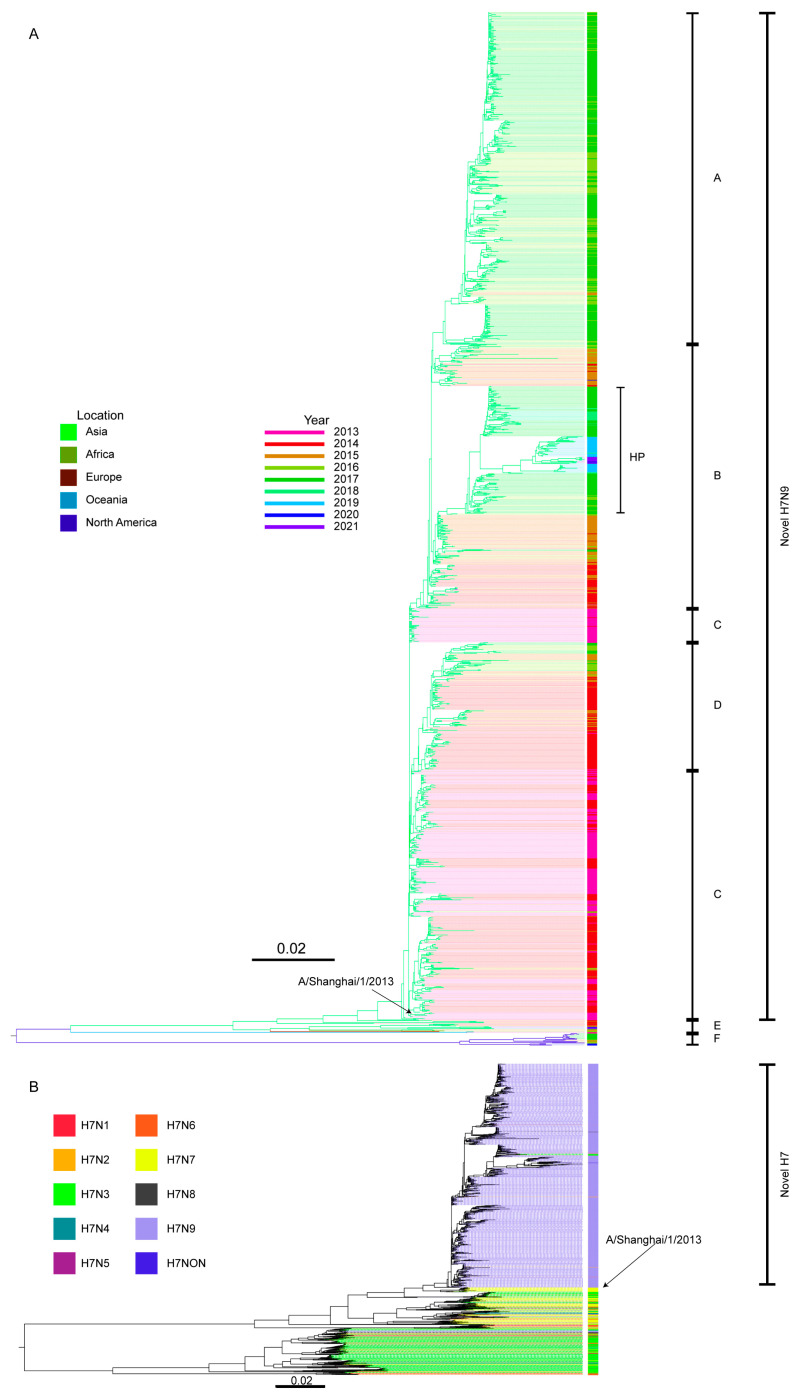
Phylogenetic analysis of the HA genes of H7 influenza viruses from 2013 to 2022. (**A**) Phylogenetic tree of the HA genes of H7N9 influenza viruses. Nucleotide sequences of 1540 full-length HA genes were analyzed. The letters A, B, C, D, E, and F indicate the lineages. HP: highly pathogenic avian influenza virus. (**B**) Phylogenetic tree of the HA genes of the H7 influenza viruses. Nucleotide sequences of 2135 full-length HAs were analyzed. Both of the phylogenetic trees are rooted at the midpoint. The trees were generated by the neighbor-joining (NJ) method in MEGA 11 software. The different branch color indicates sample collection location of virus strain. The middle color ribbon indicates the year of virus collection. Scale bar: 0.02 amino acid substitutions per site.

**Figure 3 viruses-15-02214-f003:**
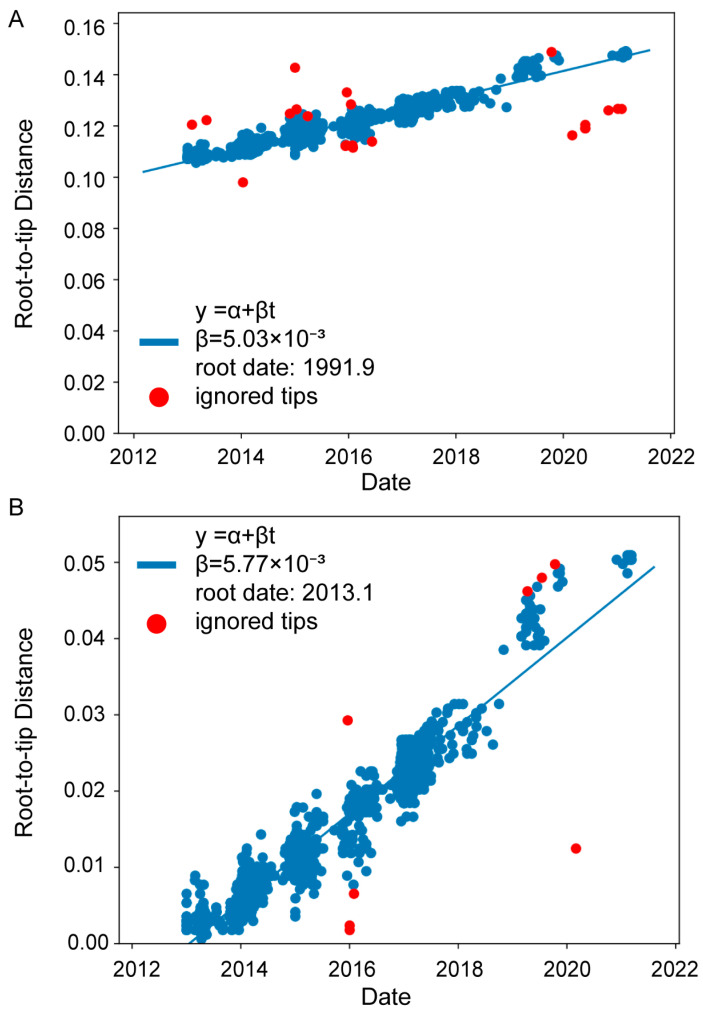
The linear relationship between the root-to-tip genetic distance and the time of virus collection. The tMRCA of the HA sequences from all the H7N9 viruses (**A**) and the novel H7N9 viruses (**B**) was estimated, respectively. The linear regression was performed using the Treetime V 0.11.1 software. *X*-axis represents the time of virus collection and *Y*-axis represents the root-to-tip genetic distance on the NJ tree. The correlation coefficient (R^2^) between the two variables is 0.95. Viruses are represented by circular points, with blue circles indicating conformity to the linear relationship and red circles representing outliers.

**Figure 4 viruses-15-02214-f004:**
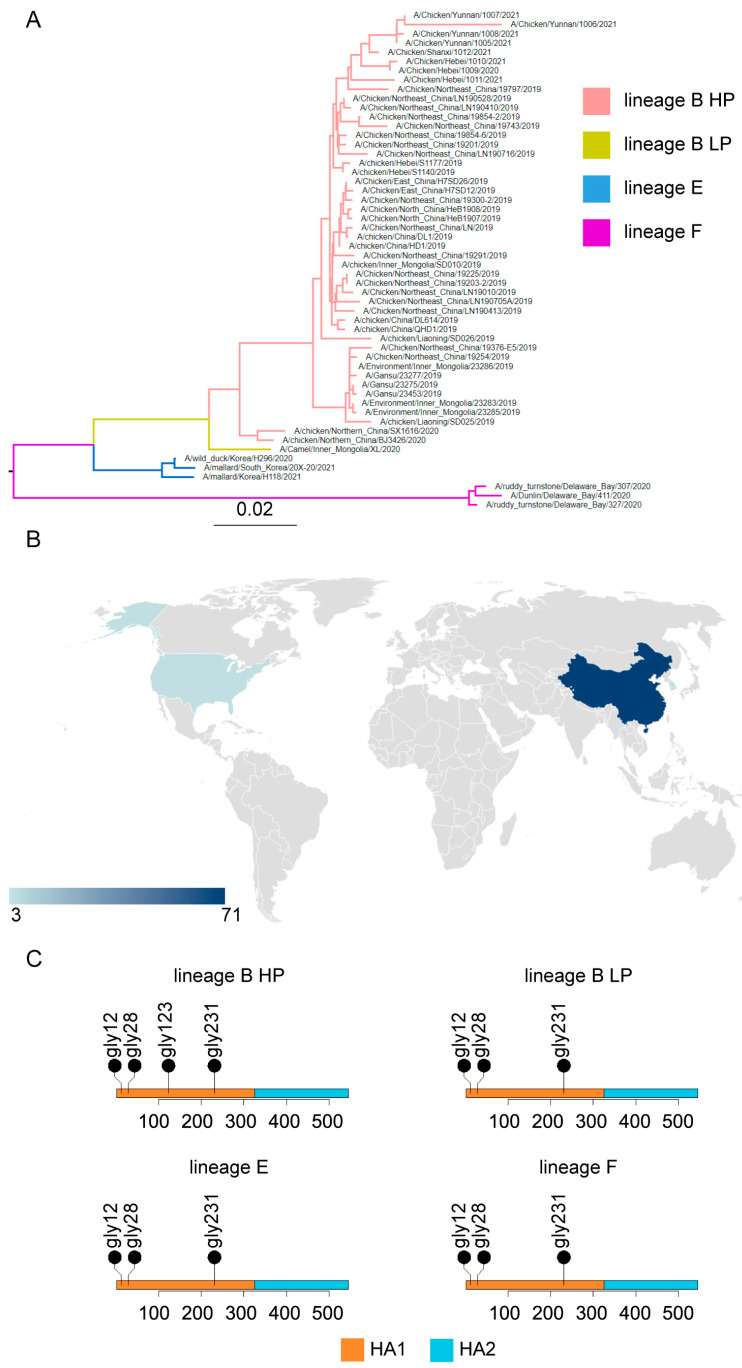
Characteristics of HA evolution, prevalence and N-glycans of H7N9 avian influenza viruses after 2019. (**A**) Phylogenetic analysis of HA amino acid sequences of the H7N9 avian influenza viruses. The tree was generated by the neighbor-joining (NJ) method in MEGA 11 software. Amino acid sequences of 54 full-length HAs were analyzed. The different branch color indicates the HA lineages. (**B**) The world map shows the total number of H7N9 viruses isolated in different countries. (**C**) Prediction of N-glycan positions on the HA proteins of H7N9 avian influenza viruses. The lollipop plot was constructed using the TrackViewer V 1.36.2 software package in the R. The HA1 and HA2 regions are distinguished using different colors.

**Figure 5 viruses-15-02214-f005:**
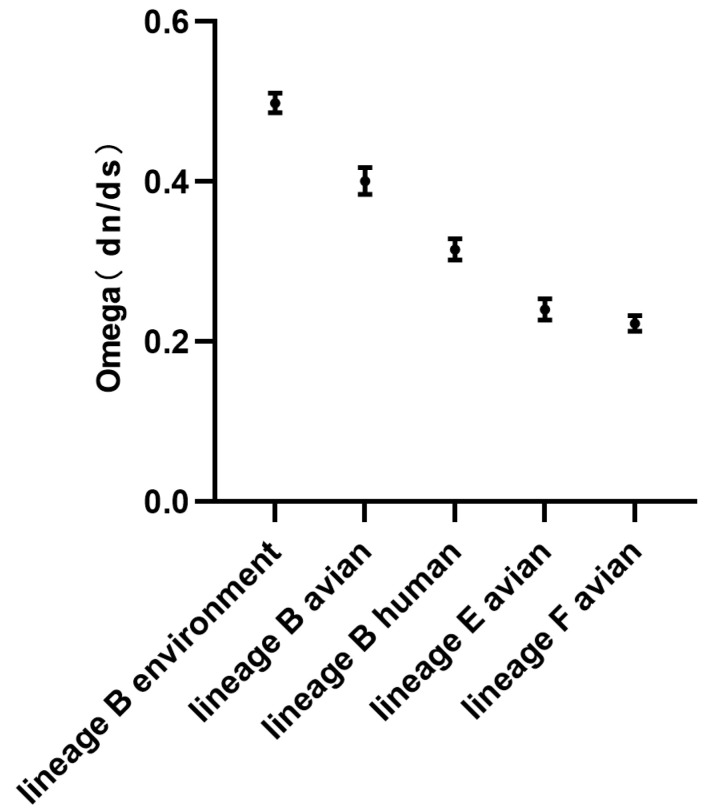
Selection pressure analysis of HA genes of the different lineages after 2019. The dots represent the average value of dn/ds calculated by the MEME and SLAC methods, with the SD values.

**Table 1 viruses-15-02214-t001:** Prediction of N-glycan positions on the HA protein.

Lineages	N-Glycan Positions on the HA Protein			
	12	28	123	231
Lineage B China HP	+++	++	++	++
Lineage B China LP	+++	++	Null	++
Lineage E Korea	++	++	Null	++
Lineage F American	++	++	Null	++

++: Potential < 0.5 and Jury agreement (9/9) OR Potential < 0.75; +++: Potential < 0.75 and Jury agreement; Null: non-glycan positions.

**Table 2 viruses-15-02214-t002:** The dn/ds values calculated by the MEME and SLAC methods.

Lineage	MEME	SLAC	Average
lineage B human	0.306	0.325	0.316
lineage B avian	0.389	0.413	0.401
lineage B environment	0.490	0.507	0.499
lineage E avian	0.231	0.250	0.241
lineage F avian	0.216	0.230	0.223

## Data Availability

The viruses’ sequence, date and location of sample collection, and other information involved in this study can all be found in GISAID.

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
