# Peer review of "Global Prevalence and Hemagglutinin Evolution of H7N9 Avian Influenza Viruses from 2013 to 2022"

_viruses, 2023, doi:10.3390/v15112214_

Round 1
Reviewer 1 Report
The manuscript by Liu et al. describes the analysis of over 2500 H7N9 viral sequences. The authors divided the sequences into six phylogenetic groups and discussed the evolutionary changes to the hemagglutinin gene over time and space. There is much value in the current manuscript, but as some critical information is missing plus the issues of language, this manuscript is not suitable for publication in its present form.
There are issues with language throughout the manuscript. I will point out some ready changes in the Minor Issues section of my comments, but a thorough revision by a native speaker is recommended.
I have concerns with the use of the term “novel” and “traditional” viral strains in this manuscript. Historically, poultry H7N9 strains have their origin in wild bird H7 viruses, often in combination with reassortment with other poultry avian influenza viruses. How does the hemaglutinin sequences used in this study compare with the larger global H7 genetic diversity?
How are the six lineages defined? Are there solid bootstrap values for them, if so please include that information in your revision.
Is there a single origin for the emergence of lineages A-D? If so a TMRCA analysis should be done on that ancestor.
Why is there no analysis of the other RNA segments in these viruses? Surely they do not carry the same cassette of internal genes.
Minor issues:
Line 14. Please replace “witnessed its” with “had”.
Line 16. Please replace “remained” with “persisted”.
Line 28. Please replace “were given birth by the recombining” with “was a result of the recombination”.
Line 40. Please replace “Thereby,” with “Thereafter the”.
Line 41. Please delete “significantly” if you are not providing a quantitative value for the reader to assess.
Line 42. Please replace “subsequently curbing” with “reducing”.
Line 49. Please replace “traditional” with “earlier”.
Line 50. Please replace “had already” with “have”.
Line 50. Please delete “Delaware,” to be consistent with only country names being used.
Line 51. Please delete “Alberta,” for the same reason.
Line 69. The sentence starting with “The Treetime…” does not belong here. it is a statement of methods.
Line 98. Please delete the first phrase and begin the sentence at “The HA gene…”
Line 102. Please replace “could be” with “was” as the M&M is written in past tense.
Line 113. Please move the first phrase to after “conducted”.
Line 135. Please delete “its” and “concentrated”.
Line 233. Please consider rephrasing this sentence. I am unclear as to its meaning.
Line 235. It is better to say: “Specific sites were analysed using Datamonkey for…
Line 259. Please replace “began evolving” with “emerged”.
Figure 2. Please make the branches darker. The different lineages are not readily differentiated in the tree provided.
Literature Cited. Some references have words in the titled capitalized and other do not. Please keep to a consistent format.
See above
Author Response
September 12, 2023
Dear Reviewer,
Thanks very much for taking your time to review this manuscript. I really appreciate all your comments and suggestions! Please find my itemized responses in below and my revisions/corrections in the re-submitted files.
Thanks again
Sincerely,
Guiqin Wang
The corresponding author
Major concerns:
- There are issues with language throughout the manuscript. I will point out some ready changes in the Minor Issues section of my comments, but a thorough revision by a native speaker is recommended.
According to the reviewer’s comment, the manuscript has been revised by a native English speaker.
The modified parts are as follows:
Line 15 18 30 42 53 54 54 73 107 112 123 146 271 274 314
- I have concerns with the use of the term “novel” and “traditional” viral strains in this manuscript. Historically, poultry H7N9 strains have their origin in wild bird H7 viruses, often in combination with reassortment with other poultry avian influenza viruses. How does the hemaglutinin sequences used in this study compare with the larger global H7 genetic diversity?
We agree with the reviewer that it is not suitable to use the term “novel” and “traditional” viral strains in this manuscript. Such descriptions of “the novel H7N9 influenza viruses” mostly appear in the literature[1-4]. The first emergence of the novel H7N9 influenza viruses was in Shanghai, China in March 2013. The novel H7N9 influenza viruses originated from a common ancestor: the virus A/Shanghai/1/2013(H7N9). The virus is a reassortant of H7, N9 and H9N2 avian influenza viruses. However, such descriptions of “the traditional H7N9 influenza viruses” has never appeared in the literature. We removed “traditional” when describing the H7N9 viruses ( line 53 56 59 60 170 173 174 181 193 201 329 331 333)
According to the reviewer’s comment, we compared the genetic diversity of the H7N9 virus hemagglutinin sequences used in this study with the global H7 virus hemagglutinin sequences. The genetic diversity of the hemagglutinin sequences was measured by the mean genetic distance. The two have a positive correlation. The mean genetic distance of the HA sequences from all the H7, H7N9 and novel H7N9 viruses from 2013 to 2022 is 0.07, 0.03 and 0.02, respectively (line208).
- How are the six lineages defined?
In this study, based on the location and time of virus isolation, the common ancestor, the evolutionary tree structure and previous studies, the HA sequence of H7N9 influenza virus was divided into 6 lineages: A, B, C, D, E and F.
The viruses belonging to lineages A, B, C, and D were classified as the novel H7N9 influenza viruses. The novel H7N9 influenza viruses include the virus A/Shanghai/1/2013(H7N9) and its descendants that have emerged in mainland China since March 2013[5]. Previous studies showed that HA sequences of the novel H7N9 viruses have been classified into two main lineages based on their origins: the Pearl River Delta lineage and the Yangtze River Delta lineage [6]. The lineage D in our study is the Pearl River Delta lineage and the remaining lineages belong to the Yangtze River Delta lineage. The viruses from the lineage D originated from the Pearl River Delta and spread to the province of Jiangsu, Hunan and Guangxi. The Yangtze River Delta lineage was subdivided into three sub-lineages: A, B, and C. The viruses from the lineages A, B and C originated from the Yangtze River Delta and spread to many provinces of China.
The viruses of lineages E and F do not belong to the novel H7N9 viruses. The viruses from the lineage F emerge exclusively in the United States. The lineage E viruses have been found in Oceania, Africa, Europe, and Asia.
- Are there solid bootstrap values for them, if so please include that information in your revision.
There were no solid bootstrap values in this study.
- Is there a single origin for the emergence of lineages A-D? If so a TMRCA analysis should be done on that ancestor.
There is a single origin for the emergence of lineages A, B, C and D. According to the reviewer’s comment The results showed that the estimated time of the most recent common ancestor (tMRCA) of the HA sequences of the novel H7N9 viruses was January 2013 and an R-squared value was 0.95 in line231([6]).
- Why is there no analysis of the other RNA segments in these viruses? Surely, they do not carry the same cassette of internal genes.
Indeed, the H7N9 viruses in our study do not carry the same cassette of internal genes. But HA serves as one of the key membrane proteins of avian influenza A viruses. The HA protein is responsible for binding the viruses to sialic acid receptors on the surface of host cells, and is one of the factors that determine the host species. The HA protein is also an important target protein for humoral immunity of influenza virus HA. The HA plays a critical role in the transmission, evolution, immune response and lifecycle of avian influenza viruses. It is crucial in preventing and controlling avian influenza pandemics to understand the genetic evolution characteristics of HA.
Minor issues:
- Some issues with language throughout the manuscript
Line 14. Please replace “witnessed its” with “had”.
Line 16. Please replace “remained” with “persisted”.
Line 28. Please replace “were given birth by the recombining” with “was a result of the recombination”.
Line 40. Please replace “Thereby,” with “Thereafter the”.
Line 41. Please delete “significantly” if you are not providing a quantitative value for the reader to assess.
Line 42. Please replace “subsequently curbing” with “reducing”.
Line 49. Please replace “traditional” with “earlier”.
Line 50. Please replace “had already” with “have”.
Line 50. Please delete “Delaware,” to be consistent with only country names being used.
Line 51. Please delete “Alberta,” for the same reason.
Line 69. The sentence starting with “The Treetime…” does not belong here. it is a statement of methods.
Line 98. Please delete the first phrase and begin the sentence at “The HA gene…”
Line 102. Please replace “could be” with “was” as the M&M is written in past tense.
Line 113. Please move the first phrase to after “conducted”.
Line 135. Please delete “its” and “concentrated”.
Line 233. Please consider rephrasing this sentence. I am unclear as to its meaning.
Line 235. It is better to say: “Specific sites were analysed using Datamonkey for…
Line 259. Please replace “began evolving” with “emerged”.
According to the reviewer’s comments we revised the language throughout the manuscript in Line 15 18 30 42 53 54 54 73 107 112 123 146 271 274 314
- Figure 2. Please make the branches darker. The different lineages are not readily differentiated in the tree provided.
According to the reviewer’s comments we made the branches darker in line212.
- Literature Cited. Some references have words in the titled capitalized and other do not. Please keep to a consistent format.
According to the reviewer’s comments we adjusted the format of some references so that the format of all the references is consistent in line 529
- Obenauer, J.C.; Denson, J.; Mehta, P.K.; Su, X.; Mukatira, S.; Finkelstein, D.B.; Xu, X.; Wang, J.; Ma, J.; Fan, Y.; et al. Large-Scale Sequence Analysis of Avian Influenza Isolates. Science 2006, 311, 1576-1580, doi:10.1126/science.1121586.
- Olson, S.; Gilbert, M.; Cheng, M.C.; Mazet, J.A.K.; Joly, D. Historical Prevalence and Distribution of Avian Influenza Virus A(H7N9) among Wild Birds. Emerging Infectious Disease journal 2013, 19, 2031, doi:10.3201/eid1912.130649.
- Cheng, M.C.; Lee, M.S.; Ho, Y.H.; Chyi, W.L.; Wang, C.H. Avian Influenza Monitoring in Migrating Birds in Taiwan During 1998–2007. Avian Diseases 2010, 54, 109-114, doi:10.1637/8960-061709-Reg.1.
- Lebarbenchon, C.; Pedersen, J.C.; Sreevatsan, S.; Ramey, A.M.; Dugan, V.G.; Halpin, R.A.; Ferro, P.J.; Lupiani, B.; Enomoto, S.; Poulson, R.L.; et al. H7N9 Influenza A Virus in Turkeys in Minnesota. Journal of General Virology 2015, 96, 269-276, doi:https://doi.org/10.1099/vir.0.067504-0.
- Gao, R.; Cao, B.; Hu, Y.; Feng, Z.; Wang, D.; Hu, W.; Chen, J.; Jie, Z.; Qiu, H.; Xu, K.; et al. Human Infection with a Novel Avian-Origin Influenza A (H7N9) Virus. The New England journal of medicine 2013, 368, 1888-1897, doi:10.1056/NEJMoa1304459.
- Zhang, J.; Ye, H.; Li, H.; Ma, K.; Qiu, W.; Chen, Y.; Qiu, Z.; Li, B.; Jia, W.; Liang, Z.; et al. Evolution and Antigenic Drift of Influenza A (H7N9) Viruses, China, 2017–2019. Emerging Infectious Disease journal 2020, 26, 1906, doi:10.3201/eid2608.200244.

Reviewer 2 Report
The authors tried to examine the evolution of avian influenza A(H7N9) viruses around the world. However, there are some problems in this study as indicated below.
Specific Remarks:
1) P.2 line 77: The official name of GISAID is incorrect. “Avian” is not needed.
2) P.2 line 78, 84 etc: The authors confuse virus isolation with sample collection. What GISAID gives us is information about the date and location of sample collection.
3) P4 line 156: The authors classified HA gene of A(H7N9) viruses into 6 lineages. Please provide reference.
4) The authors used only the H7HA genetic information of A(H7N9) viruses in this study. However, viruses with the H7HA gene often have NA genes other than N9, especially with the exception of the Chinese-H7N9 lineage that emerged in 2013.
(ref. Virus Genes 56, 472-479 (2020))
It is not suitable to consider the evolutionary rate of H7HA gene and protein only based on the H7HA information of the A(H7N9) viruses.
In fact, among the Korean viruses classified as lineage E by the authors, viruses with NA other than N9NA are classified into the same H7HA lineage.
(ref. Viruses 2021, 13(11), 2274)
5) The authors examined the virus evolution using a combination of the data from viruses of different hosts; birds, humans, and the environment.
Previous study has shown that evolutionary trends differ between viruses collected from human and that collected from birds.
(ref. Int. J. Mol. Sci. 2020, 21(19), 7129)
It is necessary to consider the data separately depending on its host.
Author Response
September 12, 2023
Dear Reviewer,
Thanks very much for taking your time to review this manuscript. I really appreciate all your comments and suggestions! Please find my itemized responses in below and my revisions/corrections in the re-submitted files.
Thanks again
Sincerely,
Guiqin Wang
The corresponding author
Specific Remarks:
- P2 line 77: The official name of GISAID is incorrect. “Avian” is not needed.
According to the reviewer’s comments“Avian” was not deleted in line 81
- P2 line 78, 84 etc: The authors confuse virus isolation with sample collection. What GISAID gives us is information about the date and location of sample collection.
According to the reviewer’s comments we corrected the mistake in line 82 89 100 108 110 168 223 229 388
- P4 line 156: The authors classified HA gene of A(H7N9) viruses into 6 lineages. Please provide reference.
In this study, according to the location and time of virus isolation, the common ancestor, the evolutionary tree structure and previous studies, the HA sequence of H7N9 influenza virus was divided into 6 lineages: A, B, C, D, E and F.
The viruses belonging to lineages A, B, C, and D were classified as the novel H7N9 influenza viruses. The novel H7N9 influenza viruses include the virus A/Shanghai/1/2013(H7N9) and its descendants that have emerged in mainland China since March 2013 [5]. Previous studies showed that HA sequences of the novel H7N9 viruses have been classified into two main lineages based on their origins: the Pearl River Delta lineage and the Yangtze River Delta lineage [6]. The lineage D in our study is the Pearl River Delta lineage and the remaining lineages belong to the Yangtze River Delta lineage. The viruses from the lineage D originated from the Pearl River Delta and spread to the province of Jiangsu, Hunan and Guangxi. The Yangtze River Delta lineage was subdivided into three sub-lineages: A, B, and C. The viruses from the lineages A, B and C originated from the Yangtze River Delta and spread to many provinces of China.
The viruses of lineages E and F do not belong to the novel H7N9 viruses. The viruses from the lineage F emerge exclusively in the United States. The lineage E viruses have been found in Oceania, Africa, Europe, and Asia.
- The authors used only the H7HA genetic information of A(H7N9) viruses in this study. However, viruses with the H7HA gene often have NA genes other than N9, especially with the exception of the Chinese-H7N9 lineage that emerged in 2013.(ref. Virus Genes 56, 472-479 (2020)). It is not suitable to consider the evolutionary rate of H7HA gene and protein only based on the H7HA information of the A(H7N9) viruses. In fact, among the Korean viruses classified as lineage E by the authors, viruses with NA other than N9NA are classified into the same H7HA lineage.(ref. Viruses 2021, 13(11), 2274)
We agree with the reviewer that it is not suitable to only analyzing the H7N9 HA sequences in this manuscript. We added genetic evolution analysis of the H7HA sequences from 2013 to 2022 in line 212
- The authors examined the virus evolution using a combination of the data from viruses of different hosts; birds, humans, and the environment. Previous study has shown that evolutionary trends differ between viruses collected from human and that collected from birds.(ref. Int. J. Mol. Sci. 2020, 21(19), 7129).It is necessary to consider the data separately depending on its host.
We agree with the reviewer that it is not suitable to analyze the selection pressure regardless of host factors. Previous studies showed that Homo sapiens have a strong selection pressure on viruses compared to chickens. The selection pressure was analyzed according to different virus hosts, and dN/dS was calculated respectively in line 280
- Gao, R.; Cao, B.; Hu, Y.; Feng, Z.; Wang, D.; Hu, W.; Chen, J.; Jie, Z.; Qiu, H.; Xu, K.; et al. Human Infection with a Novel Avian-Origin Influenza A (H7N9) Virus. The New England journal of medicine 2013, 368, 1888-1897, doi:10.1056/NEJMoa1304459.
- Zhang, J.; Ye, H.; Li, H.; Ma, K.; Qiu, W.; Chen, Y.; Qiu, Z.; Li, B.; Jia, W.; Liang, Z.; et al. Evolution and Antigenic Drift of Influenza A (H7N9) Viruses, China, 2017–2019. Emerging Infectious Disease journal 2020, 26, 1906, doi:10.3201/eid2608.200244.

Reviewer 3 Report
Line 52: I understand the sensitivity of this issue but consider listing Taiwan with ownership. It currently reads "Taiwan, China".....I would suggest changing to "Taiwan".
Lines 78-80: why exclude any virus with the word reassortant in the name? Suspected reassorted viruses may have been included in your data set.
Lines 41-49: Vaccination of poultry is mentioned here and then forgotten. I was hoping that the impact of vaccination on viral evolution would appear in the results or the discussion but does not. In the Discussion, can you please address the impact of vaccination on viral evolution? Can you compare the evolution of viruses collected from areas that did not employ vaccination to those that came from areas that did?
General comments:
AIV contains a segmented genome. The fact that the H and N genes are not linked is not presented at all. They do not need to evolve at the same rate.
That being said, the analysis in the manuscript relies on data collected by others. It is not clear how the consensus sequences were constructed/determined at time of publication to the GISAID database. The potential impact to the analysis presented in this manuscript should be addressed somewhere in the discussion.
Although the viruses were broken down by host origin, it would also be interesting to break them down by source.....digestive tract sample, respiratory tract sample, other. The environmental samples were clearly digestive tract/fecal samples......be it direct fecal sample or water filtration. This may impact the evolutionary analysis.
Author Response
October 9, 2023
Dear Reviewer 3:
Thanks very much for taking your time to review this manuscript(viruses-2565080). I really appreciate all your comments and suggestions! Please find my itemized responses in below and my revisions in the re-submitted files.
- Line 52: I understand the sensitivity of this issue but consider listing Taiwan with ownership. It currently reads "Taiwan, China".....I would suggest changing to "Taiwan".
Answer: We agree with the reviewer and revised our description of Taiwan
- Lines 78-80: why exclude any virus with the word reassortant in the name? Suspected reassorted viruses may have been included in your data set.
Answer: From 2013 to 2022, there are a total of 8 strains of recombinant virus in the GISAID database , which are:
A/Shanghai/HK/6:2/2013 (reassortant 6:2 A/Hong Kong/1/1968/162/35 x A/Shanghai/2/2013-PR8-IDCDC),
A/reassortant/IDCDC-RG56N(A/Hong Kong/125/2017 X Puerto Rico/8/1934),
A/reassortant/IDCDC-RG56B (A/Hong Kong/125/2017 X Puerto Rico/8/1934),
A/reassortant/NIBRG268(Anhui/1/2013 x Puerto Rico/8/1934),
A/reassortant/NIBRG268(A/Anhui/1/2013 x Puerto Rico/8/1934),
A/Anhui/01/2013-CDC-LV7A and A/Anhui/1/2013 NIIDRG-10.1.
The HAs of these recombinant viruses are the same as those of wild virus strains, which have been included in our study.
- Lines 41-49: Vaccination of poultry is mentioned here and then forgotten. I was hoping that the impact of vaccination on viral evolution would appear in the results or the discussion but does not. In the Discussion, can you please address the impact of vaccination on viral evolution? Can you compare the evolution of viruses collected from areas that did not employ vaccination to those that came from areas that did?
Answer: According to the reviewer’s comment, we added a paragraph to the discussion section about the effect of vaccination on the prevalence and evolution of the novel H7N9 influenza(line 336).
The novel H7N9 influenza viruses caused five waves of human infections from 2013 to 2017, with a total of 1568 cases, of which 615 were fatal[1]. The novel H7N9 influenza viruses were low pathogenicity in birds. In early 2017 some low pathogenic H7N9 strains became highly pathogenic in chickens, causing severe influenza outbreaks in poultry in China[1]. H7N9 viruses posed the severe threat to human and animal health. To control the epidemic, the poultry were mandatorily immunized with H7N9 inactivated vaccine from September 2017 in China[2]. After birds were inoculated with the H7 vaccine, infection cases in humans and birds dramatically reduced. This suggests that the national poultry vaccination strategy has successfully controlled H7N9 infections in poultry and humans. Compared to the earlier novel H7N9 viruses, the HA gene of the H7N9 virus from the latest human case showed a relatively long genetic distance[1]. This indicates that the H7N9 viruses have evolved considerably even though vaccination dramatically limited the H7N9 prevalence. H7N9 viruses will not be eradicated. The viruses lurk in natural hosts such as waterfowls and continue to evolve, occasionally infecting other animals such as poultry. Once the virus has accumulated enough mutations and recombination, it could lead to the next pandemic. Based on World Health Organization (WHO) recommendation, a total of 6 virus strains were used for the development of influenza A(H7N9) vaccines from 2013 to 2023. The WHO may continue to recommend vaccine strains until all cases of human and poultry infection have disappeared.
Figure: Human infections with H7N9 viruses in China from 2013 to 2022
Table: H7N9 viruses isolated between February 2017 and January 2018
The figure is cited from Alarming situation of emerging H5 and H7 avian influenza and effectivecontrol strategies[2].
The table is cited from Rapid evolution of H7N9 highly pathogenic viruses that emerged in China in 2017 [3].
- Can you compare the evolution of viruses collected from areas that did not employ vaccination to those that came from areas that did?
Answer: We compared the HA evolution of viruses collected from areas that did not employ vaccination to those that came from vaccination areas in Figure 5.
- AIV contains a segmented genome. The fact that the H and N genes are not linked is not presented at all. They do not need to evolve at the same rate. That being said, the analysis in the manuscript relies on data collected by others. It is not clear how the consensus sequences were constructed/determined at time of publication to the GISAID database. The potential impact to the analysis presented in this manuscript should be addressed somewhere in the discussion.
Answer: We agree with the reviewer. In the discussion section, we added the description of the delay in uploading the collected virus data to the GISAID database(line357).
- Although the viruses were broken down by host origin, it would also be interesting to break them down by source.....digestive tract sample, respiratory tract sample, other. The environmental samples were clearly digestive tract/fecal samples......be it direct fecal sample or water filtration. This may impact the evolutionary analysis.
Answer: Virus samples from other sources showed different evolutionary characteristics compared to samples from the environment. The DN/DS ratio is larger (Figure 5), which indicates that these samples undergone more positive selection pressure and less negative selection pressure(line276)
Sincerely,
Guiqin Wang
The corresponding author
References
- Li, C.; Chen, H. H7n9 Influenza Virus in China. Cold Spring Harb Perspect Med 2021, 11, doi:10.1101/cshperspect.a038349.
- Shi, J.; Zeng, X.; Cui, P.; Yan, C.; Chen, H. Alarming Situation of Emerging H5 and H7 Avian Influenza and Effective Control Strategies. Emerging Microbes & Infections 2023, 12, 2155072, doi:10.1080/22221751.2022.2155072.
- Shi, J.; Deng, G.; Ma, S.; Zeng, X.; Yin, X.; Li, M.; Zhang, B.; Cui, P.; Chen, Y.; Yang, H.; et al. Rapid Evolution of H7n9 Highly Pathogenic Viruses That Emerged in China in 2017. Cell Host & Microbe 2018, 24, 558-568.e557, doi:https://doi.org/10.1016/j.chom.2018.08.006.

Reviewer 4 Report
This article is very important both for science and for practical point of view. However, some major and minor points need to be clarified.
Point 1: As part of the WHO Global Influenza Programme, twice a year (for the southern and northern hemisphere) since at least 2010, regular recommendations have been made on the composition of influenza vaccines against zoonotic (avian) influenza (https://www/who.int/teams/global-influenza-programme/vaccines/who-recommendations/zoonotic-influenza-viruses-and-candidate-vaccine-viruses). Please indicate when WHO recommendations for the development of H7N9 vaccines were first made. Most likely, it was about the A/Anhui/1/2013 virus. Discuss situation as of today and the last WHO meeting of 29 September 2023, when, for the first time in recent years, no recommendations were made on zoonotic vaccines. During the period from February 2023, sporadic or single cases of H5N6, H3N8, H9N2, H5N1, H1N1v and H1N2v were reported. According WHO H7N9 were not detected at all. Please, discuss – why it happened. I realize that your study covers a 10-year time period ending in December 2022, but it would be very helpful to discuss the latest information from WHO.
Point 2: According to the figure legends, Figure 1A should present a global prevalence of H7N9 avian influenza from 2013 to 2022. In fact, the results end in December 2020. Please, add data on H7N9 distribution in 2021 and 2022. Of course, the authors indicated that “Since March 2021, H7N9 viruses have not been isolated.” Nevertheless, for completeness these time points should be added to the Figure.
Point 3: Since 2017 only sporadic cases of H7N9 viruses were detected around the world. Can the authors scientifically predict when the next wave of H7N9 avian influenza viruses will enter circulation? Will the absence of these viruses today match their prediction?
Point 4: Again, according to Figure 1 only sporadic cases of H7N9 viruses were detected around the world since 2017. However, in 2017-2022 among other vaccines against potentially dangerous avian influenza H7N9 viruses WHO recommended also H7N9 and only at the very last WHO meeting recommendations for H7N9 viruses were not made. Could you explain – why?
Point 5: The authors presented very interesting data on the evolution of H7N9 viruses, but don’t you think that the almost complete absence of H7N9 viruses in circulation over the past 5-6 years, including today, suggests that these viruses have exhausted themselves? Should we expect the emergence of fundamentally new reassortant avian viruses in the near future and how to prepare for this?
Point 6: Line 10. I would replace “For the purpose of further understanding” with “For further understanding”
Point 7: Line 24. Please, replace “select pressure” with “selective pressure”
Point 8: Line 89. I would replace “For the purpose of reducing” with “For reducing”
Point 9: Line 161-162. Please, add reference after the statement “HA genes were classified into six distinct lineages labeled A, B, C, D, E, and F.”
Author Response
October 9, 2023
Dear Reviewer 4:
Thanks very much for taking your time to review this manuscript(viruses-2565080). I really appreciate all your comments and suggestions! Please find my itemized responses in below and my revisions in the re-submitted files.
- Point 1: As part of the WHO Global Influenza Programme, twice a year (for the southern and northern hemisphere) since at least 2010, regular recommendations have been made on the composition of influenza vaccines against zoonotic (avian) influenza (https://www/who.int/teams/global-influenza-programme/vaccines/who-recommendations/zoonotic-influenza-viruses-and-candidate-vaccine-viruses). Please indicate when WHO recommendations for the development of H7N9 vaccines were first made. Most likely, it was about the A/Anhui/1/2013 virus. Discuss situation as of today and the last WHO meeting of 29 September 2023, when, for the first time in recent years, no recommendations were made on zoonotic vaccines. During the period from February 2023, sporadic or single cases of H5N6, H3N8, H9N2, H5N1, H1N1v and H1N2v were reported. According WHO H7N9 were not detected at all. Please, discuss – why it happened. I realize that your study covers a 10-year time period ending in December 2022, but it would be very helpful to discuss the latest information from WHO.
Answer: Based on genetic and antigenic analysis, it is recommended that: An A/Anhui/1/2013-like* virus be used for the development of influenza A(H7N9) vaccines for pandemic preparedness purposes in March 2013. From 2013 to 2023, a total of 6 virus strains were used for the development of influenza A(H7N9) vaccines (table 2). The WHO may continue to recommend vaccine strains until all cases of human and poultry infection have disappeared (line336).
Date |
Candidate vaccine virus |
References |
2023.10 |
A/Guangdong/17SF003/2016; A/HongKong/125/2017; A/Shanghai/2/2013; A/Anhui/1/2013; A/Gansu/23277/2019-like; A/Hunan/02650/2016 |
[1] |
2023.2 |
A/Guangdong/17SF003/2016; A/Hong Kong/125/2017; A/Shanghai/2/2013; A/Anhui/1/2013; A/Gansu/23277/2019-like; A/Hunan/02650/2016 |
[2] |
2022.9 |
A/Guangdong/17SF003/2016; A/Hong Kong/125/2017; A/Shanghai/2/2013; A/Anhui/1/2013; A/Gansu/23277/2019-like; A/Hunan/02650/2016 |
[3] |
2022.2 |
A/Guangdong/17SF003/2016; A/Hong Kong/125/2017; A/Shanghai/2/2013; A/Anhui/1/2013; A/Gansu/23277/2019-like; A/Hunan/02650/2016 |
[4] |
2021.9 |
A/Guangdong/17SF003/2016; A/Hong Kong/125/2017; A/Shanghai/2/2013; A/Anhui/1/2013; A/Gansu/23277/2019-like; A/Hunan/02650/2016 |
[5] |
2021.3 |
A/Guangdong/17SF003/2016; A/Hong Kong/125/2017; A/Shanghai/2/2013; A/Anhui/1/2013; A/Gansu/23277/2019-like; A/Hunan/02650/2016 |
[6] |
2020.9 |
A/Guangdong/17SF003/2016; A/Hong Kong/125/2017; A/Shanghai/2/2013; A/Anhui/1/2013; A/Gansu/23277/2019-like; A/Hunan/02650/2016 |
[7] |
2020.2 |
A/Guangdong/17SF003/2016; A/Hong Kong/125/2017; A/Shanghai/2/2013; A/Anhui/1/2013; A/Gansu/23277/2019-like; A/Hunan/02650/2016 |
[8] |
2019.9 |
A/Guangdong/17SF003/2016; A/Hong Kong/125/2017; A/Shanghai/2/2013; A/Anhui/1/2013; A/Gansu/23277/2019-like; A/Hunan/02650/2016 |
[9] |
2019.2 |
A/Guangdong/17SF003/2016 A/Hong Kong/125/2017 A/Shanghai/2/2013 A/Anhui/1/2013 A/Hunan/02650/2016 |
[10] |
2018.11 |
A/Guangdong/17SF003/2016 A/Hong Kong/125/2017 A/Shanghai/2/2013 A/Anhui/1/2013 A/Hunan/02650/2016 |
[11] |
2018.3 |
A/Guangdong/17SF003/2016 A/Hong Kong/125/2017 A/Shanghai/2/2013 A/Anhui/1/2013 A/Hunan/02650/2016 |
[12] |
2017.11 |
A/Guangdong/17SF003/2016 A/Hong Kong/125/2017 A/Shanghai/2/2013 A/Anhui/1/2013 |
[13] |
2017.5 |
A/Hong Kong/125/2017 A/Shanghai/2/2013 A/Anhui/1/2013 |
[14] |
2016.2 |
A/Shanghai/2/2013 A/Anhui/1/2013 |
[15] |
2015.3 |
A/Shanghai/2/2013 A/Anhui/1/2013 |
[16] |
2014.2 |
A/Shanghai/2/2013 A/Anhui/1/2013 |
[17] |
2013.12 |
A/Shanghai/2/2013 A/Anhui/1/2013 |
[18] |
2013.8 |
A/Shanghai/2/2013 A/Anhui/1/2013 |
[19] |
- Point 2: According to the figure legends, Figure 1A should present a global prevalence of H7N9 avian influenza from 2013 to 2022. In fact, the results end in December 2020. Please, add data on H7N9 distribution in 2021 and 2022. Of course, the authors indicated that “Since March 2021, H7N9 viruses have not been isolated.” Nevertheless, for completeness these time points should be added to the Figure.
Answer: According to the reviewer’s comment, the manuscript has been revised in Figure 1A.
- Point 3: Since 2017 only sporadic cases of H7N9 viruses were detected around the world. Can the authors scientifically predict when the next wave of H7N9 avian influenza viruses will enter circulation? Will the absence of these viruses today match their prediction?
Answer: H7N9 viruses will not be eradicated. The viruses lurk in natural hosts such as waterfowls and continue to evolve, occasionally infecting other animals such as poultry. Once the virus has accumulated enough mutations and recombination, it could lead to the next pandemic.
- Point 4: Again, according to Figure 1 only sporadic cases of H7N9 viruses were detected around the world since 2017. However, in 2017-2022 among other vaccines against potentially dangerous avian influenza H7N9 viruses WHO recommended also H7N9 and only at the very last WHO meeting recommendations for H7N9 viruses were not made. Could you explain – why?
Answer: From 2013 to 2023, a total of 6 virus strains were used for the development of influenza A(H7N9) vaccines (table 2). The WHO may continue to recommend vaccine strains until all cases of human and poultry infection disappear (line336).
Date |
Candidate vaccine virus |
References |
2023.10 |
A/Guangdong/17SF003/2016; A/HongKong/125/2017; A/Shanghai/2/2013; A/Anhui/1/2013; A/Gansu/23277/2019-like; A/Hunan/02650/2016 |
[1] |
2023.2 |
A/Guangdong/17SF003/2016; A/Hong Kong/125/2017; A/Shanghai/2/2013; A/Anhui/1/2013; A/Gansu/23277/2019-like; A/Hunan/02650/2016 |
[2] |
2022.9 |
A/Guangdong/17SF003/2016; A/Hong Kong/125/2017; A/Shanghai/2/2013; A/Anhui/1/2013; A/Gansu/23277/2019-like; A/Hunan/02650/2016 |
[3] |
2022.2 |
A/Guangdong/17SF003/2016; A/Hong Kong/125/2017; A/Shanghai/2/2013; A/Anhui/1/2013; A/Gansu/23277/2019-like; A/Hunan/02650/2016 |
[4] |
2021.9 |
A/Guangdong/17SF003/2016; A/Hong Kong/125/2017; A/Shanghai/2/2013; A/Anhui/1/2013; A/Gansu/23277/2019-like; A/Hunan/02650/2016 |
[5] |
2021.3 |
A/Guangdong/17SF003/2016; A/Hong Kong/125/2017; A/Shanghai/2/2013; A/Anhui/1/2013; A/Gansu/23277/2019-like; A/Hunan/02650/2016 |
[6] |
2020.9 |
A/Guangdong/17SF003/2016; A/Hong Kong/125/2017; A/Shanghai/2/2013; A/Anhui/1/2013; A/Gansu/23277/2019-like; A/Hunan/02650/2016 |
[7] |
2020.2 |
A/Guangdong/17SF003/2016; A/Hong Kong/125/2017; A/Shanghai/2/2013; A/Anhui/1/2013; A/Gansu/23277/2019-like; A/Hunan/02650/2016 |
[8] |
2019.9 |
A/Guangdong/17SF003/2016; A/Hong Kong/125/2017; A/Shanghai/2/2013; A/Anhui/1/2013; A/Gansu/23277/2019-like; A/Hunan/02650/2016 |
[9] |
2019.2 |
A/Guangdong/17SF003/2016 A/Hong Kong/125/2017 A/Shanghai/2/2013 A/Anhui/1/2013 A/Hunan/02650/2016 |
[10] |
2018.11 |
A/Guangdong/17SF003/2016 A/Hong Kong/125/2017 A/Shanghai/2/2013 A/Anhui/1/2013 A/Hunan/02650/2016 |
[11] |
2018.3 |
A/Guangdong/17SF003/2016 A/Hong Kong/125/2017 A/Shanghai/2/2013 A/Anhui/1/2013 A/Hunan/02650/2016 |
[12] |
2017.11 |
A/Guangdong/17SF003/2016 A/Hong Kong/125/2017 A/Shanghai/2/2013 A/Anhui/1/2013 |
[13] |
2017.5 |
A/Hong Kong/125/2017 A/Shanghai/2/2013 A/Anhui/1/2013 |
[14] |
2016.2 |
A/Shanghai/2/2013 A/Anhui/1/2013 |
[15] |
2015.3 |
A/Shanghai/2/2013 A/Anhui/1/2013 |
[16] |
2014.2 |
A/Shanghai/2/2013 A/Anhui/1/2013 |
[17] |
2013.12 |
A/Shanghai/2/2013 A/Anhui/1/2013 |
[18] |
2013.8 |
A/Shanghai/2/2013 A/Anhui/1/2013 |
[19] |
Point 5: The authors presented very interesting data on the evolution of H7N9 viruses, but don’t you think that the almost complete absence of H7N9 viruses in circulation over the past 5-6 years, including today, suggests that these viruses have exhausted themselves? Should we expect the emergence of fundamentally new reassortant avian viruses in the near future and how to prepare for this?
Answer: To control the epidemic, the poultry were mandatorily immunized with H7N9 inactivated vaccine from September 2017 in China. After birds were inoculated with the H7 vaccine, infection cases in humans and birds dramatically reduced. This suggests that the national poultry vaccination strategy has successfully controlled H7N9 infections in poultry and humans. But the HA gene of the H7N9 virus from the latest human case showed a relatively long genetic distance. This indicates that the H7N9 viruses have evolved considerably even though vaccination dramatically limited the H7N9 prevalence. H7N9 viruses will not be eradicated. The viruses lurk in natural hosts. Once the virus has accumulated enough mutations and recombination, it could lead to the next pandemic. So, the WHO and the virologists worldwide are actively preparing for the next pandemic of H7N9 influenza. The WHO and may continue to recommend vaccine strains until all cases of human and poultry infection have disappeared. The vaccinologists are developing the universal vaccine against the next influenza pandemic. The next influenza pandemic could be H7N9, H1N1,H5N1 and so on. Universal influenza vaccines would be one of the most effective measures to prevent and control an influenza pandemic.
Point 6: Line 10. I would replace “For the purpose of further understanding” with “For further understanding”
Point 7: Line 24. Please, replace “select pressure” with “selective pressure”
Point 8: Line 89. I would replace “For the purpose of reducing” with “For reducing”
Answer: According to the reviewer’s comment, the manuscript has been revised in line10,24 and 89.
Point 9: Line 161-162. Please, add reference after the statement “HA genes were classified into six distinct lineages labeled A, B, C, D, E, and F.”
Answer: In this study, based on the location and time of virus isolation, the common ancestor, the evolutionary tree structure and previous studies, the HA sequence of H7N9 influenza virus was divided into 6 lineages: A, B, C, D, E and F.
The viruses belonging to lineages A, B, C, and D were classified as the novel H7N9 influenza viruses. The novel H7N9 influenza viruses include the virus A/Shanghai/1/2013(H7N9) and its descendants that have emerged in mainland China since March 2013[23]. Previous studies showed that HA sequences of the novel H7N9 viruses have been classified into two main lineages based on their origins: the Pearl River Delta lineage and the Yangtze River Delta lineage [24]. The lineage D in our study is the Pearl River Delta lineage and the remaining lineages belong to the Yangtze River Delta lineage. The viruses from the lineage D originated from the Pearl River Delta and spread to the province of Jiangsu, Hunan and Guangxi. The Yangtze River Delta lineage was subdivided into three sub-lineages: A, B, and C. The viruses from the lineages A, B and C originated from the Yangtze River Delta and spread to many provinces of China.
The viruses of lineages E and F do not belong to the novel H7N9 viruses. The viruses from the lineage F emerge exclusively in the United States. The lineage E viruses have been found in Oceania, Africa, Europe, and Asia.
Sincerely,
Guiqin Wang
The corresponding author
Reference
- WHO. Summary of Status of Development and Availability of Avian Influenza a(H7n9) Candidate Vaccine Viruses and Potency Testing Reagents. Available online: https://www.who.int/publications/m/item/a(h7n9)---southern-hemisphere-2024 (accessed on 2023.10).
- WHO. Summary of Status of Development and Availability of Avian Influenza a(H7n9) Candidate Vaccine Viruses and Potency Testing Reagents. Available online: https://www.who.int/publications/m/item/a(h7n9)---northern-hemisphere-2023-2024 (accessed on 2023.2).
- WHO. Summary of Status of Development and Availability of Avian Influenza a(H7n9) Candidate Vaccine Viruses and Potency Testing Reagents. Available online: https://cdn.who.int/media/docs/default-source/influenza/cvvs/cvv-zoonotic---southern-hemisphere-2023/summary_a_h7n9_cvv_20220928.pdf?sfvrsn=401f5216_4 (accessed on 2022.9).
- WHO. Summary of Status of Development and Availability of Avian Influenza a(H7n9) Candidate Vaccine Viruses and Potency Testing Reagents. Available online: https://cdn.who.int/media/docs/default-source/influenza/cvvs/cvv-zoonotic---northern-hemisphere-2022-2023/h7n9_summary_a_h7n9_cvv_20220225.pdf?sfvrsn=67f60ee5_9 (accessed on 2022.2).
- WHO. Summary of Status of Development and Availability of Avian Influenza a(H7n9) Candidate Vaccine Viruses and Potency Testing Reagents. Available online: https://cdn.who.int/media/docs/default-source/influenza/cvvs/cvv-zoonotic---southern-hemisphere-2022/h7n9_summary_a_h7n9_cvv_20210930.pdf?sfvrsn=a9939e84_8 (accessed on 2021.9).
- WHO. Summary of Status of Development and Availability of Avian Influenza a(H7n9) Candidate Vaccine Viruses and Potency Testing Reagents. Available online: https://cdn.who.int/media/docs/default-source/influenza/cvvs/cvv-zoonotic---northern-hemisphere-2021-2022/summary_a_h7n9_cvv_nh2021_22_20210305.pdf?sfvrsn=ccc739b8_14 (accessed on 2023.3).
- WHO. Summary of Status of Development and Availability of Avian Influenza a(H7n9) Candidate Vaccine Viruses and Potency Testing Reagents. Available online: https://cdn.who.int/media/docs/default-source/influenza/cvvs/cvv-zoonotic---southern-hemisphere-2021/summary_a_h7n9_cvv_sh2021_20201001.pdf?sfvrsn=6ea5febd_18 (accessed on 2020.9).
- WHO. Summary of Status of Development and Availability of Avian Influenza a(H7n9) Candidate Vaccine Viruses and Potency Testing Reagents. Available online: https://cdn.who.int/media/docs/default-source/influenza/cvvs/cvv-zoonotic---northern-hemisphere-2020-2021/summary-a-h7n9-cvv-nh2020-21-20200228.pdf?sfvrsn=a2bc5a75_11 (accessed on 2020.2).
- WHO. Summary of Status of Development and Availability of Avian Influenza a(H7n9) Candidate Vaccine Viruses and Potency Testing Reagents Available online: https://cdn.who.int/media/docs/default-source/influenza/cvvs/cvv-zoonotic---southern-hemisphere-2019/summary-a-h7n9-cvv-sh2020-20190926.pdf?sfvrsn=f40ece8f_11 (accessed on 2019.9).
- WHO. Summary of Status of Development and Availability of Avian Influenza a(H7n9) Candidate Vaccine Viruses and Potency Testing Reagents. Available online: https://cdn.who.int/media/docs/default-source/influenza/cvvs/cvv-zoonotic---northern-hemisphere-2019-2020/summary-a-h7n9-cvv-nh1920-20190220.pdf?sfvrsn=5b177d5d_11 (accessed on 2019.2).
- WHO. Summary of Status of Development and Availability of Avian Influenza a(H7n9) Candidate Vaccine Viruses and Potency Testing Reagents. Available online: https://cdn.who.int/media/docs/default-source/influenza/cvvs/cvv-zoonotic---southern-hemisphere-2018/summary-a-h7n9-cvv-20181108.pdf?sfvrsn=559becac_11 (accessed on 2018.11).
- WHO. Summary of Status of Development and Availability of Avian Influenza a(H7n9) Candidate Vaccine Viruses and Potency Testing Reagents. Available online: https://cdn.who.int/media/docs/default-source/influenza/cvvs/cvv-zoonotic---northern-hemisphere-2018-2019/summary-a-h7n9-cvv-20180808.pdf?sfvrsn=80768449_11 (accessed on 2018.3).
- WHO. Summary of Status of Development and Availability of Avian Influenza a(H7n9) Candidate Vaccine Viruses and Potency Testing Reagents. Available online: https://cdn.who.int/media/docs/default-source/influenza/cvvs/cvv-zoonotic---southern-hemisphere-2017/summary_a_h7n9_cvv_20171114.pdf?sfvrsn=c68f8320_11 (accessed on 2017.11).
- WHO. Summary of Status of Development and Availability of Avian Influenza a(H7n9) Candidate Vaccine Viruses and Potency Testing Reagents. Available online: https://cdn.who.int/media/docs/default-source/influenza/cvvs/cvv-zoonotic---northern-hemisphere-2017-2018/summary_a_h7n9_cvv_20170518.pdf?sfvrsn=2978e6f2_11 (accessed on 2017.5).
- WHO. Summary of Status of Development and Availability of Avian Influenza a(H7n9) Candidate Vaccine Viruses and Potency Testing Reagents. Available online: https://cdn.who.int/media/docs/default-source/influenza/cvvs/cvv-zoonotic---northern-hemisphere-2016-2017/summary_a_h7n9_cvv_20160229.pdf?sfvrsn=c2e4fb4_13 (accessed on
- WHO. Summary of Status of Development and Availability of Avian Influenza a(H7n9) Candidate Vaccine Viruses and Potency Testing Reagents. Available online: https://cdn.who.int/media/docs/default-source/influenza/cvvs/cvv-zoonotic---northern-hemisphere-2015-2016/summary_a_h7n9_cvv_20150317.pdf?sfvrsn=28e6cf75_11 (accessed on
- WHO. Summary of Status of Development and Availability of Avian Influenza a(H7n9) Candidate Vaccine Viruses and Potency Testing Reagents. Available online: https://cdn.who.int/media/docs/default-source/influenza/cvvs/cvv-zoonotic---northern-hemisphere-2014-2015/summary_a_h7n9_cvv_20140213.pdf?sfvrsn=d0618776_12 (accessed on
- WHO. Summary of Status of Development and Availability of Avian Influenza a(H7n9) Candidate Vaccine Viruses. Available online: https://cdn.who.int/media/docs/default-source/influenza/cvvs/cvv-zoonotic---northern-hemisphere-2013-2014/summary_a_h7n9_cvv_20130813.pdf?sfvrsn=700ac604_12 (accessed on
- WHO. Summary of Status of Development and Availability of Avian Influenza a(H7n9) Candidate Vaccine Viruses 2013.

Round 2
Reviewer 1 Report
Thank you for addressing the recommendations by the reviewers. This version of the manuscript is much better. There are still a number of places, especially in the Abstract and Introduction sections where the writing should be addressed but once they are, this manuscript should be ready for publication.
Line 11. Please replace “drew the” with “generated a”.
Line 14. Please delete ‘also”. Please delete “periodic”.
Line 17. Please relace “being the” with “becoming”.
Line 18. You data show that Lineage B has only one additional predicted glycosylation site. Please replace “with a higher number of” with “has an additional predicted”, and replace “suffer” with “and is under”.
Line 31. Please delete “the” in front of “H9N2 viruses”, and delete “the” in front of “novel”.
Line 70. Please delete “the” in front of “H7N9”.
Line 89. Please replace “Subsequently, this” with “Our”.
Line 254. Please insert a space after “lineage F”.
Line 317. Please replace “By” with “Using” or “Utilizing”.
Line 423. Please delete “Fortunately,”.
Figure 4. Lineage E. The glycosylation site at 231 is mislabeled as 123.
Author Response
October 9, 2023
Dear Reviewer 1:
Thanks very much for taking your time to review this manuscript(viruses-2565080). I really appreciate all your comments and suggestions! Please find my itemized responses in below and my revisions in the re-submitted files.
Minor issues:
- Comments on the Quality of English Language
Line 11. Please replace “drew the” with “generated a”.
Line 14. Please delete ‘also”. Please delete “periodic”.
Line 17. Please relace “being the” with “becoming”.
Line 18. You data show that Lineage B has only one additional predicted glycosylation site. Please replace “with a higher number of” with “has an additional predicted”, and replace “suffer” with “and is under”.
Line 31. Please delete “the” in front of “H9N2 viruses”, and delete “the” in front of “novel”.
Line 70. Please delete “the” in front of “H7N9”.
Line 89. Please replace “Subsequently, this” with “Our”.
Line 254. Please insert a space after “lineage F”.
Line 317. Please replace “By” with “Using” or “Utilizing”.
Line 423. Please delete “Fortunately,”.
Answer: According to the reviewer’s comment, the manuscript has been revised in line11,14,17,18,31,57,69,170,211 and 288.
- Figure 4. Lineage E. The glycosylation site at 231 is mislabeled as 123.
Answer: According to the reviewer’s comment we corrected the error in Figure 4.
Sincerely,
Guiqin Wang
The corresponding author

Reviewer 2 Report
The second version of the paper still has many problems. The author’s purpose in this paper is fundamentally flawed, and the meaning of each analysis should be reconsidered.
The authors still analyzed the evolution of H7HA using data from only A(H7N9) viruses (Fig.3A, Fig 4, Table 1, Fig.5, Table 2). Avian influenza viruses with the H7HA gene, with the exception of the Chinese-H7N9 lineage that emerged in 2013, are often confirmed to have reassorted and have NA genes other than N9 (Fig 2B). Therefore, it was a serious mistake to analyze the evolution of H7HA of lineage E and F classified by the authors, which is generally classified into Eurasian and North American lineages, using only data from A(H7N9) viruses.
Furthermore, the authors’ analysis methods are misinterpreted in their paper, such as confusing the evolution of genes and amino acids, and focusing on selection pressure of viruses over a short period of time.
The authors should think carefully about the purpose of the research and conduct analysis that is appropriate for it.

Author Response
October 9, 2023
Dear Reviewer2:
The research object of this paper is the global H7N9 virus from 2013 to 2022, so the novel and non-novel H7N9 viruses are analyzed together.
Sincerely,
Guiqin Wang
The corresponding author
